# Design of Wireless Sensors for IoT with Energy Storage and Communication Channel Heterogeneity

**DOI:** 10.3390/s19153364

**Published:** 2019-07-31

**Authors:** Paul Nicolae Borza, Mihai Machedon-Pisu, Felix Hamza-Lup

**Affiliations:** 1Department of Electronics and Computers, Faculty of Electrical Engineering and Computers, Transilvania University of Brasov, 500024 Brasov, Romania; 2Department of Computer Science, Georgia Southern University, P.O. Box 7997, Stateboro, GA 30460, USA

**Keywords:** wireless sensor nodes, autonomous sensors, electric energy storage, spectrum coexistence, energy management, internet of things

## Abstract

Autonomous Wireless Sensors (AWSs) are at the core of every Wireless Sensor Network (WSN). Current AWS technology allows the development of many IoT-based applications, ranging from military to bioengineering and from industry to education. The energy optimization of AWSs depends mainly on: Structural, functional, and application specifications. The holistic design methodology addresses all the factors mentioned above. In this sense, we propose an original solution based on a novel architecture that duplicates the transceivers and also the power source using a hybrid storage system. By identifying the consumption needs of the transceivers, an appropriate methodology for sizing and controlling the power flow for the power source is proposed. The paper emphasizes the fusion between information, communication, and energy consumption of the AWS in terms of spectrum information through a set of transceiver testing scenarios, identifying the main factors that influence the sensor node design and their inter-dependencies. Optimization of the system considers all these factors obtaining an energy efficient AWS, paving the way towards autonomous sensors by adding an energy harvesting element to them.

## 1. Introduction

In the current context of the Internet of Things (IoT), the possibility to develop smart and context aware applications in different environments (rural, urban, and industrial) is a reality enabled by autonomous wireless sensors (AWS). Autonomous wireless sensors represent the core of the Wireless Sensor Networks (WSN). They must deliver data with high reliability, must exhibit high energetic performance as well as autonomy. Current technology allows the development of a large spectrum of sensor-based applications in various fields, from military to bioengineering and from industry to education. The increased complexity of the sensors’ behavior raises new challenges regarding reliability, availability, accuracy, energy consumption, security, and data transfer efficiency, in an extremely complex environment. Such complexity spawns the development of simulation test beds that facilitate the decision process in the hardware and software design of the next generation AWSs. For each AWS parameter, it is important to identify its variation range and the cross correlation with other parameters. The AWS architecture refers to components and their organization, illustrating the subsystem inferences. Thus, in the case of power sources, the storage system should simultaneously satisfy the application demands: Both for basic power and short-term power variation. The informational aspect will mainly influence the transceiver choice and the protocol with which is endowed. In Figure 1, the main components of the optimization process are revealed.

The paper commences with a survey identifying the AWS’ main parameters, their ranges, and effects on energy efficiency. In this sense, the following steps are considered: Definition and adoption of an appropriate structure and topology for the WSN;Decision on the parameters that must be optimized from the energetic perspective;Development of an AWS prototype in order to simulate, using real components, several use cases and highlight the relationship between data and energy consumption in accordance with the application’s requirements (e.g., spectrum and storage system life span).

The proposed methodology for sizing and controlling of the AWS hybrid power supply (battery combined with super-capacitor) was developed, implemented, and tested. A wide variety of sensors technologies are available on the market today. The principal research question is related to the “hybridization” methodology of the autonomous wireless sensors. The premises for structural and functional hybridization of the main subsystems of the AWS are analyzed, including the power management strategies in order to improve the sensors’ autonomy. Two aspects were considered: The power flow optimization between the storage elements (battery and super-capacitor) and the power management related to the wireless transmission and the successive functional stages of the micro-controller and sensor transceiver. In accordance with the proposed architectural changes for the AWS (i.e., their hardware and corresponding software solutions) were implemented. Additionally, we analyze how these changes lead to saving resources, increasing the AWS system’ life-time, and also an improving functionality. The AWS parameters (e.g., the field of emission and the data exchanged) must be correlated with the energy consumption and transceiver type in order to provide recommendations for sensor parameter adjustment (e.g., sensor placement, power tuning, communication protocols, and transceiver types or other hardware combinations).

The paper is structured as follows: in Section 2 we present a survey of transceiver technologies for AWSs and their associated parameters, as well as the powering methods for AWS. Section 3 provides the transceiver’s testing methodology followed in Section 4 by the AWS design and implementation. In Section 5 we discuss the open field AWS testing results followed in Section 6 by a 3D visualization of EM fields and AWS’ power consumption. In Section 7, we propose the hybridization of the communication by including dual transceivers and energy sources for improved AWS behavior. A methodology is proposed for sizing the hybrid supply in order to avoid accelerated-aging, by minimizing the stress produced by instantaneous parameters, such as fast current and voltage variation, on battery. Section 8 describes two IoT-based applications. Improvements in transceiver performance may occur by increasing the power density of the power supply, i.e., reducing the equivalent series resistance of the AWS power source, as shown herein.

## 2. Related Work

Wireless Sensor Networks (WSNs) provide the “cells” for data collection and distribution within IoT enabling the development of smart, context aware applications. By sharing multiple types of power sources and maintaining power autonomy for large periods of time, these devices are the real enablers of the IoT, in terms of lifetime, energy efficiency, low costs, and connectivity. Moreover, advances in electric energy storage systems have pushed sensor autonomy to new levels. 

### 2.1. Transceivers, Standards and Parameters

A wide range of WSN standards for communication for short, medium, and long range exist, implemented on a variety of communication protocols and various frequencies, using a wide range of power sources, as summarized in Table 1 [1,2,3,4,5,6,7]. However, not all of them can be applied to real-world applications, where factors such as: environmental conditions (temperature, humidity, etc.), radio spectrum performance (antenna design and band coexistence), and energy storage (rechargeable batteries, energy harvesting) play a crucial role. Moreover, autonomy puts additional constraints specifically on the sensor’s powering methods. 

When choosing the components of an AWS, the following aspects must be considered:Communication protocol. Energy consumption, latency and throughput for different Medium Access Control (MAC) protocols for WSNs may have a significant impact on the sensor’s performance [8]. A significant reduction in energy consumption (i.e., 18%–45%) was obtained for MAC protocols based on Bluetooth (BT) nodes with increased throughput and lower latency. Experimental data proves that Bluetooth Low Energy (BLE) is more energy efficient when compared to the ZigBee protocol. Translated in power consumption, an improvement from 35–40 mW to 12–16 mW can be gained, as illustrated in Table 1. The possibility to develop smart applications with BLE is reviewed in Reference [9]. Solutions based on BLE are more efficient than the Wi-Fi based implementations. Comparing Wi-Fi with BLE in terms of power consumption, Table 1 and Table 2 illustrate BLE’s advantages.Components cost. While most commercial devices for WSNs are expensive and proprietary, and as IoT continues to grow, more resources are needed for building smart WSNs with lower costs. The performance of built-for-purpose devices against open-source devices is analyzed in Reference [10]. Based on the analysis, the most expensive proprietary devices for WSNs are based on the ZigBee standard.Sensor lifetime. BLE can increase the lifetime of the system for up to 5 years in some cases [11]. Recently, novel BLE mesh topology with improved scalability, sustainability, and coverage was explored [12]. A systematic review of BLE’s performance and limitations is presented in Reference [13]. Unfortunately, studies on network coverage and energy consumption for different operations or models that follow real-world power consumption based on bit rate and topology variations are absent.WSN topology. A different topology may be employed for achieving optimal performance, when attenuation and interference sources are present. The star topology is based on peer-to-peer communications among the gateway and the WSNs. Hybrid and mesh topology are more adaptable to the environment’s radio settings and nodes failure, by enabling new density of network nodes.Range. The BT/BLE transceivers have a short range compared to RFM transceivers that work at more than several hundred meters. In case of ZigBee and Wi-Fi, there are medium range transceivers. Wi-Fi consumes high energy when communicating at the range limits.Communication reliability. An important aspect, less investigated in the research literature, is the reliability of the WSN in terms of transceiver antenna and band coexistence. BLE based transceivers allow a much shorter range, gain, and sensitivity threshold than ZigBee and Wi-Fi, as illustrated in Table 1. It is possible to use a directional antenna instead of an omnidirectional one, commonly implemented by ZigBee and Wi-Fi [14]. The benefits are: Improved energy efficiency, transmission range, and fewer collisions. The coexistence in the 2.4 GHz band is still controversial, especially between ZigBee and Wi-Fi [15].Security. WSNs communicate sensitive data, thus security concerns must be addressed at the beginning of the system design [16,17,18,19,20,21,22,23]. The main aspects deal with: Limited resources [16], unreliable communication [16,17], unattended operation [16], data integrity and confidentiality [18,19], authentication [19], time synchronization [19], secure localization [19], traffic analysis attacks [20], and countermeasures to attacks [21], like cryptography and key establishment [22,23]. Due to the resource, space, and cost constraints placed on the sensor nodes in a WSN [24], many of the traditional security solutions are not suitable. The large number of threats makes it very difficult to build security solutions for WSNs.Application requirements. WSNs are used in many domains, e.g., military, industrial, environmental, residential, and health care [25,26]. Applications include smart homes (systems based on own Wi-Fi platforms [27], or commercial: ESP8266 [28,29]) to smart cities (including smart transportation [30], smart governance [31,32], and smart grid [32])smart utilities(especially water [33,34,35] and energy management [33,35] systems) to smart cars (including software defined networks [36], automotive applications [37], smart parking systems based on ZigBee platforms [38], and car security-based on Arduino Uno board [39]), and precision agriculture (mainly smart farming and irrigation with Wi-Fi platforms, such as ESP8266 [40] and ZigBee platforms, such as eZ430 [41] or 3G/4G/Wi-Fi connections [42] to e-health solutions (mainly patient monitoring and support with Raspberry Pi board [43], or with ZigBee platforms, such as Xbee [44], or with Bluetooth [45]). Depending on their requirements and sensor capabilities, one can define WSNs in terms of size (small to very large scale), sensors’ capacity (homogeneous to heterogeneous), topology, and mobility (static, mobile, and hybrid) [46]. Many types of WSN architectures are presented in literature, such as these: Based on DAQ boards [47], for indoor localization [48], based on intelligent gateways [49], industrial [50], and global/ heterogeneous sensor data networks [51]. IoT-based architectures for WSNs are reviewed in Reference [52]. A flexible architecture can be achieved, as discussed inReference [53]. All applications can benefit from new, low-power WSN standards and platforms, as illustrated in References [47,48,49,50,51]. By taking them into account, a modular IoT architecture is proposed in Reference [4]. While LoRa and ZigBee [48,50,51] are perceived as more suitable, most implementations do not consider, in their analysis, IoT-based requirements such as connectivity and cost (illustrated in the last columns of Table 1 and Table 2). Different WSN deployment strategies can be adapted in this sense to solve coverage, network connectivity, deployment cost, energy efficiency, life span, data fidelity, and load balancing issues. The cost of Zigbee solutions, especially Xbee-based, is still high enough for low-cost IoT implementations. On the other hand, Lora has adopted a very efficient modulation, respectively, chirp spread spectrum modulation for achieving low power, simultaneously increasing the range. At the same time, this protocol shows a higher robustness to interference. The costs for transceivers are kept low and are able to support high data rates. Mentioned features make this protocol very attractive for implementing a large spectrum of IoT applications [54,55].

No proprietary solution for the WSNs can fully address all requirements. The possibility to develop BLE mesh networks with both academic and proprietary solutions have been proposed [56]. Others [57] do not consider power management options such as energy harvesting and recharging cycles, nor the heterogeneous nature of the autonomous sensor nodes. The data in Table 2 proves this point of view in terms of coverage and power consumption.

### 2.2. Energy Sources and Storage for AWS

The ever-increasing demand for wireless services comes at the price of a considerable electromagnetic and carbon foot print of the wireless communications industry. Minimizing this footprint is a challenge that is seldom addressed, however there is a strong economic driver to reduce the energy consumption of the wireless networks. As predicted by several analysts [58], mobile data traffic is increasing dramatically every year, essentially due to a major increase in mobile video traffic.

The increase in the overall network energy consumption is driven by the computational complexity of advanced transmission techniques, the increasing number of base stations required for high data rates and electromagnetic pollution. As the energy prices are increasing, it becomes obvious that the trade-off between radio performance and energy efficiency will become extremely important for near future WSNs.

The energy requirements are even more stringent for WSN with autonomous nodes. The powering method has important consequences, assuring the continuity of functionality between two consecutive battery re-charging sessions. There are four solutions for powering the AWS:Battery. The specificity of WSN-related applications requires the use of energy sources that have to meet constraints such as: Being mechanically robust, having high energy/power densities, and exceptional lifespan. Recent developments in micro batteries are related to the development of controlled 3D atomic structures that generate exceptional properties and high performance [59]. LiPO (Lithium Polymer) batteries, or other new implementation like NiSn-LMO (Nickel tin-anode, Lithiated Manganese Oxide-cathode) reach ~440 Whkg^−1^. In the case of Li-air batteries, the energy density is higher and can reach 700 Whkg^−1^ [60]. These values are comparable with the liquid fuel energy density. Despite the batteries technological progress, two main issues still remain: The relatively high internal resistance and reduced cycle-ability and life span of batteries.Super-capacitors (SC). Therecent evolution of the SC domain shows a significant extension of the temperature domain (−40 °C at more than +150 °C), in parallel with an increase in capacity (more than 550 Fg^−1^ theoretical value, at huge specific surface more than 2675 m^2^g^−1^), power (10 Wg^−1^), and energy density (more than 10 mWhg^−1^) comparable with Li-Ion batteries 100 mWhg^−1^). The significant increase of energy density at values similar to lead-acid batteries, make these solutions very attractive for future developments. An actual manifested trend illustrates the research and development of a fully integrated solid-state device that merges transceivers and storage elements on the same system.Hybrid Energy Storage System (HESS)—as a combination of batteries and SC. In this case, the high-power density of SC will be in accordance with the transceiver needs. Moreover, hybrid SCs have one electrode based on Faradaic phenomena (chemical), and a second one based on non-Faradaic phenomena (electrostatic).Harvesting-based Systemsassure an infinite life span for the AWSs, if the harvesting generator is properly integrated with the storage element. The sizing of the storage element must shadow the attributes of the energy harvesting system (e.g., solar, mechanical, thermal, electromagnetic, or piezoelectric). Various AWSs were proposed, employing ZigBee and energy harvesting mechanisms [58,59,60,61,62], however, these implementations not only lack detailed lifetime analysis based on environment and spectrum information, but also lack relevant cost estimations. Additionally, we show that it is crucial to perform analyses based on the energy consumed over a transmitted bit so as to precisely determine the impact of operation phases on the wireless transceiver consumption. A similar solution with a harvesting system consists of building a WSN that uses both wireless transfer of signal (information) and also energy. This solution is investigated in Reference [63]. In References [64,65], various mathematical models are proposed as topology and organization of WSNs are redesigned [66,67]. For improved autonomy, the strict control of the AWS’s energy state becomes of crucial importance. Current trends propose the replacement of classic batteries with new storage solutions (e.g., micro super-capacitors) that present many advantages (e.g., weight, extended temperature domain, life span, robustness, power, and energy density).

The maintenance costs of the AWS are affected by the energy storage system (ESS). An appropriate design of ESS or HESS can reduce or entirely eliminate the need to service a specific AWS. HESS or harvesting based solutions extend the charging cycle of the AWS with beneficial consequences for the system’s autonomy. For energy harvesting systems, the size of the sensor’s ESS will influence the AWS’s life span. The majority of the ESSs used for AWSs are based on carbon materials, having different allotropic forms: Graphene, graphite, or a combination of carbon with other elements like vanadium oxide, titanium, lithium, sodium, or potassium. Many of these storage cells use a solid electrolyte that makes these very reliable, robust, and compact [59].

It is also possible to consider sharing the computational load between homogeneous AWSs (ZigBee, Wi-Fi, BT, and RF) and the WSN sink [4], improving the energy consumption by balancing the computational load, however a discussion of these methods is beyond the scope of this paper.

## 3. Transceiver Testing Methodology

Considering the crucial importance of transceivers used on AWS, we have tested both the transceivers and the AWS prototype, separately and integrated on the designed AWS. The AWS implementation has included measurement facilities for each AWS transceivers’ consumption (BT/BLE and RFM, separately, in and outside the laboratory).We consider the transceiver’ test separately from AWS as being essential for revealing how the different wireless sensor parameters like distance, propagation environment, data flow and data payload, and the communication protocol influences the power consumption. The laboratory test bench was setup including an electric current sensor placed into the AWS’s transceiver circuit, as illustrated in Figure 2. The current was monitored by a 16 bits data acquisition card. The communication scenarios assumed transmission as the echo of data between the mobile sensor and the wireless data-collector, and unidirectional communication between the same nodes. The information transferred was chosen in order to reach the extreme (i.e., minimal and maximal) numbers of voltage transitions corresponding to data payload (00H for minimal, respectively, 55H for maximal).

The complex design process of the AWSN due to constraints (e.g., temperature, humidity, bandwidth interference, size, and power consumption) is facilitated by the cooperative work of several categories of systems designers, engineers, architects, and other decision makers that are not always collocated.

## 4. AWS Design and Implementation

Considering the actual stage of wireless technologies development, as well as the implementations mentioned in Section 2, we implemented an AWS capable of supporting a wide range of transceivers. The custom implementation allows built-in measurements of the power consumption in real-time for the corresponding transceivers. The sampling rate can be set in the range of 10 sps (samples per second) to over 10 ksps per channel, allowing real-time energy consumption measurements. The implementation enables monitoring of the power consumption under different parameters, e.g., power, sensitivity, and distance between nodes, antenna orientation, transceiver’s type, and communication protocol. The parameters that we monitor through the simulation can be classified as:Parameters associated with the transceiver performance (e.g., range, band, and power consumption).Parameters that describe the energy stored as well as the static and dynamic performances.Parameters inter-related with the communication protocol.Parameters influenced by the sensor’s physical placement and environmental conditions.

### 4.1. Hardware Implementation

The hardware layout is illustrated in Figure 3. An AVR8 family micro-controller [68,69,70,71,72,73] is used for coordination of data acquisition and control of the AWS activities including the sampling control of the telegrams for ADC. For each transceiver, the working current is measured by inserting into the supplying circuits from battery current sensors based on the Hall Effect [71]. These offer a proportional signal to the current absorbed by the AWS’s transceivers. The sensors’ working frequency is 50 kHz with a precision of 0.2%. An analogue multiplexer integrated on the AWS controller is used to monitor the current signal from the supplying circuits of the two AWS transceivers. A secondary ADC having a 16 bits resolution and a sampling rate of 15 sps is connected by an I2 C interface to the AWS controller [72]. The auto-calibrated 16 bits ADC assures the high accuracy of the converted signal under a wide range of environmental conditions. This allows the acquisition of additional signals with high resolution and low sampling rate. The temperature and humidity sensor sampling rate are functions of the signal acquired (14 bits for temperature and 12 bits for relative humidity). The input domain for temperature is from −40 °C to 125 °C and for the relative humidity from 0% and 100%.

In order to investigate the potential advantages of communication hybridization used for AWS sensors, we implemented the system using 2 transceivers (BLE and RFM—2.4 GHz). The proposed solution for AWS aims to minimize the consumption and also to extend the range and number of potential AWSs in the network. The two transceivers used, enable short (up to 10–30 m) and medium (400–1000 m) range wireless connections. These functions are controlled by the main processor of the AWS using an UART and a SPI serial interface.

The AWSs have an ATMega88 micro-controller with 8 kB flash memory and 1 kB SRAM. The sampling period for temperature and humidity can be adjusted by a corresponding command between 1 ms to more than 60 s. The firmware allows integration into network, reliable transfer of data, and the management of energy and data. Based on the most current data acquired from the current sensors, a dynamic management of energy with specific rules can be implemented.

This energy management protects and improves the lifespan of the ESS associated with the AWS. The implementation of the AWS is illustrated in Figure 4.

### 4.2. Software Components

The firmware is conceived as “an event driven” software, minimizing the interrupt service routines duration together with the usage of the CPU time [68,69,70,72,73]. The communication protocol includes a set of telegrams allowing the modification of the sampling period and the resolution for the acquired signals. Using telegrams, the set of data and the number of samples used to calculate the mean values can be adjusted. The data is ASCII encoded and can be stored into a text file on the data collector. The following functionality is implemented:Connecting and transferring data to another AWS that has similar interfaces, respectively, BT (BLE) and RFM (2.4 GHz-24L01) interfaces.Preprocessing of the acquired data: Mean values calculation, histogram of data acquired, and conversion from binary to ASCII in order to improve the telegram transfer visibility.Data transfer initialization through the chosen transceiver, as well as triggering the current acquisition signals of the transceivers on the AWS. The recording time is limited by the microcontroller memory (i.e.,8 KB).Offline transfer of the data files with the recorded currents through a serial interface at the initiative of the network data collector (UART).Allows star and mesh topology implementations.Scalable, flexible and re-configurable routines allowing quick modification of the initial setup of each AWS node.A limited set of ASCII commands transmitted through the UART serial interface, ensures system control during experiments.

## 5. Band Coexistence for Short to Medium Range Communication

Numerous studies have revealed the impact of competing wireless technologies in the same frequency bands. The cross technology interference is mostly in the ISM band: 2.4–2.48 GHz and refers to Bluetooth, ZigBee and Wi-Fi: whether it is the effect of Wi-Fi on ZigBee [74,75], on Bluetooth [76,77,78,79,80], or on both [80,81,82]. Different band coexistence solutions were proposed, including: positioning [76,77], mathematical models, such as Markov [75,81], cross-technology design [75], coordination schemes [78], and opportunistic antenna utilization [79]. Additionally, as new BT standards are proposed: BTv4.0 [83] and BTv5.0 [84], band coexistence analysis should also be addressed.

The test scenarios refer to the following real-life environments in which the propagation conditions and in-band interference were evaluated: (1) Medium Range—a good propagation (outdoor) environment with almost free spectrum (illustrated in Figure 5a), and (2) Short Range—an indoor environment with spectrum mostly occupied by Wi-Fi sources, see Figure 5b.

In the first scenario, the free spectrum can be observed for the entire ISM band: [2400–2480 MHz], which is almost at noise level (around −100 dBm)In the second scenario, the spectrum is occupied: The best case is for [2400–2420 MHz], and the worst case for [2430–2450 MHz], as seen in Figure 6a,b where Wi-Fi interference is less visible.

The impact of the environment’s propagation conditions was tested on BT/BLE communications via HC-05 and JDY-30 transceivers. The transmission distance is different in the two scenarios illustrated in Figure 6b: 30 to 40 m for the first, 10 to 15 m for the second, but this was obtained at the same floor, while between floors; the distance is only 8 m. The attenuation and reflections due to obstacles are more severe in the second scenario.

In scenarios similar to Figure 6c, we performed spectral analysis for BT/BLE transmissions with JDY-30 and HC-05 transceivers for reduced Wi-Fi interference and at a close distance to the HyperLOG 6080 logarithm antenna (around 0.5 m) connected to a real-time spectrum analyzer (Tektronix SA 2600). The spectral analysis reveals how much the spectrum is affected by external interference (band interference), the transmission power of the wireless technologies present in band, and the distance between transmission bands (free bands), all of which can lead to band coexistence solutions. One way to avoid ISM band interference is to operate in different ranges. NRF24L01 transmitters can provide this solution by transmitting in the [2480–2525 MHz] free band. As shown in Figure 6d, this band is almost free of interference (−95 dBm in the second scenario).

Few applications implement nRF transceivers, and even fewer compared to BT/BLE, ZigBee and Wi-Fi transceivers, yet the ones that do [85,86] reveal new advantages such as: Mesh networking, reliability, long-range operation [85], and lower current consumption [86]. In Table 3, we have also considered nRF24 for long-range communications. As illustrated in Reference [80], by providing a gain of 20–30 dB for the additional long-range antenna, the BER can be improved from 0.1–1 to 10^−3^–10^−4^, a 1000-fold reduction in BER. Additionally, in Table 3, the in-band interference in the outdoor scenario is negligible for most cases (with less than 20% loss in throughput).

By allowing NRF24L01 transmitters to operate outside the ISM band, dual transmitter operation is possible: BT/BLE via HC-05, JDY-30, and HM-10 can operate in the ISM band as long as Wi-Fi interference is at an acceptable level (as in the first scenario, and only in the band [2400–2420 MHz] for the second scenario, see Figure 6a,c for comparison), while RFM via NRF24L01 operates outside the ISM band in order to solve the coexistence problem in the case of internal interference. As observed for the band coexistence of BT/BLE transmissions, either the transmission power is increased (by means of an additional supply) or the spectrum is freed from external interference caused mainly by Wi-Fi sources, as in the second scenario, in Figure 5b. BT/BLE uses FHSS (frequency hopping) in order to find a free band in the spectrum, as seen in the [2400–2420 MHz] band, in the second scenario (see Figure 6c).

## 6. 3D Visualization of the AWS Emission Fields and Power Consumption

If the AWS position/orientation is known, one can compute and simulate a static 3D map of the sensor’s parameters (e.g., field of emission, field of reception, etc.). For this analysis, we consider two prominent standards for developing AWS: ZigBee, via MicaZ transceivers, and Bluetooth, via HC-05 transceivers. The propagation characteristics and RF radiation pattern of MicaZ nodes, based on monopole antennas with omnidirectional radiation pattern, are discussed in References [87,88,89,90]. Regarding Bluetooth, different antenna designs were proposed and analyzed in References [91,92,93,94]. As shown in References [95,96,97,98], different micro-strip antenna designs, most of them patch antennas, provide a directional radiation pattern. The practical measurements for the radiation pattern (emission fields) are compared to the theoretical models [99], and the results are comparable, as shown in the Table 4.

### 6.1. 3D Representation for Power Consumption Evaluation—Static Systems

The AWS’s field of emission/reception is one of the most important parameters that, even though invisible, has a strong impact on the communication quality and power consumption. Imagine being able to visualize in 3D and observe the field of emission/reception for each AWS as well as the complex intersection of their emission/reception fields. This would allow a better understanding of the interference at each AWS node level and the relationship with the power consumption as well as the architecture of the building/city neighborhood, enabling smart decisions to be taken in order to improve the overall efficiency of the entire city.

Based on laboratory experiments, a Matlab 3D representation from practical measurements (performed at a distance of 5 m in open space) is provided in Figure 7, illustrating the volume of the emission fields (in dB) for a ZigBee (MicaZ) omnidirectional (monopole) antenna and a BT/BLE (HC-05) directional (patch) antenna:

The practical measurements confirm the proximity to the theoretical model, however the small differences between the measured and the theoretical model can only be measured visually, by looking at the 3D shapes (Figure 8 illustrates the theoretical model). The plane for MicaZ antenna is OY-OZ and the radiation field plane is OX-OZ (perpendicular to it). The plane for HC-05 antenna is OX-OY and the radiation field plane is OX-OZ.

AWS’s power consumption is influenced by several factors (e.g., bandwidth collisions, interference, weather electrostatic conditions, communication protocol, data type, etc.). The 3D visualization enables a better understanding of the complex interplay among several sensors’ parameters and the observation of the sensor field shape and behavior under various functioning conditions.

The power consumption analysis of the sensors improves the AWS placement such that the sensor autonomy is improved. In this case, we expect sensors to provide data without interruption for time frames of up to one year.

### 6.2. Current Drawn by Medium to Short Range Transceivers

As in the previous chapter, we consider only medium to short-range communications. In our experimental use-case we transmit characters with “echo” successively several times (50, 200, and 500 times). The intention was to highlight the session consumption in each case and also the energy consumed for transmitting one character. As a reference, we have considered the pattern provided by TI in AN092 [100]. Figure 9, Figure 10 and Figure 11 reveal that both HC-05 and JDY-30 transceivers have a large current consumption when compared to 15–20 mA currents, according to TI [100].

We have experimentally obtained currents from 40 to 65 mA, as illustrated in Figure 9, Figure 10, Figure 11 and Figure 12, for periods of maximum transceiver activity (see Table 1 and Table 2 and [100] for comparison). Such power-hungry transceivers require additional power supply, corresponding circuits, and strategies to smooth out the charging/discharging characteristic. In the connected state, different data payloads were transmitted: From 50 to 500 characters (different commands for a number of U characters were sent in burst: 50×U cmd, 100×U cmd, 200×U cmd, 300×U cmd, and 500×U cmd). In this analysis, we also covered the connected state without any data transmissions (no command was sent = no cmd) and the disconnected state, illustrated in Figure 9, Figure 10 and Figure 11. It is also important to mention that all measurements were performed in no echo mode.

Considering the form revealed inReference [100], the values obtained in similar conditions are presented in Table 5 and Table 6. For HC-05 and also for JDY-30 commutation spikes could be observed. These significantly affect the mean values of the current consumed by the transceivers (illustrated in Figure 9, Figure 10 and Figure 11).

Figure 12 illustrates the echo mode, which means that the character received is immediately transmitted back to the sender. Related to data payload, the experiments reveal a slight difference in consumption when we transmit “U” characters (55H), respectively, null characters (00H). Figure 13, Figure 14, Figure 15 and Figure 16 depict the waveform for current consumption when transmitting a number of characters in burst: 50× U or null vs. 100× U or null characters with HC-05 and JDY-30 transceivers, which use BT 2.1, respectively 3.0.

The spikes observed for these transceivers are relevant for the total consumption measured in our experimental settlements. The differences are illustrated in Figure 17 and Figure 18. The tests are summarized in Table 7 for the echo mode.

We observe and prove that there is a dependency between the distance, the data pattern and the power consumption, that is essential for an optimal adaptation to the application or process based on wireless communication, as seen in Figure 19. For this, we chose free field conditions without significant electromagnetic field interference, as illustrated in Figure 5a.

Numerous studies have also tackled the inter-dependencies between SNR, BER, and modulation schemes with distance and for different throughput. Reference [101] analyzes the impact of different modulation schemes on BER for Bluetooth with and without FHSS. Reference [102] proposes an algorithm for estimating the SNR based on the power spectrum that can adapt to different Bluetooth packets. Reference [103] analyzes the effects of interference on both BER and throughput. The new standard for Bluetooth, BLE is discussed in Reference [104] in terms of energy consumption for different modulation schemes, emphasizing the importance of energy per bit analysis for selecting the optimal range and rate. As shown and discussed in References [101,102,103,104], these inter-dependencies have a great impact on the power consumption of the sensor node with Bluetooth transceiver. The first strategy to obtain a lower BER is to use the lowest rate modulation schemes. The second strategy is to increase the SNR. Out of these strategies, only the first assures optimal power consumption without investing in new resources, since the latter requires more powerful antennas for a better range, as in the case nRF24 [85,86], which of course leads to more power consumption. In this paper, the data rates selected for all the Bluetooth transceivers tested are at minimum: 1 Mbps for HC-05, 1 Mbps for JDY-30, and respectively, 721 kbps for HM-10. This allows data transmission at maximum range with no loss in BER.

### 6.3. Power Consumption for Medium to Short Range Communications

Three different stages were considered for the power consumption experiments with three transceivers:HM-10 (BLE), JDY-30 (BT), and HC-05 (BT) (illustrated in Figure 2):Before the pairing stage,Transmission/reception (echo mode) of a character with minimum transition stages (the ‘Null’ character 00H) andTransmission/reception (echo mode) of a character with maximum transition stages (‘U’ character, 55H).

The analogue signal was acquired with an NI6215 data acquisition card. The voltage reference was established at 0.2 V in order to obtain a maximum absolute resolution of 4.8 μV; the sampling rate for the signal acquisition was 250 ksps. During the experiments in open air, the distance between the transceiver and its corresponding system ranged between 0.3 m and 30 m. The value of the series resistance used for acquiring the transceiver current is 12.22 Ω; the stabilized power supply is V_S_ = 3.3 V or 5 V.

The transceiver’s current sink is given by the formula:(1)$$i=\frac{v\left(t\right)}{R}=\frac{v\left(t\right)}{12.22}$$

The effective voltage applied on the transceiver is:(2)$$v_{TRS}=V_{S}-v\left(t\right)$$

The power consumed by the transceiver is:(3)$$p{\left(t\right)}_{TRS}=i\cdot v_{TRS}=i\cdot \left(V_{S}-v\left(t\right)\right)=\frac{v\left(t\right)\cdot V_{S}}{12.22}-\frac{v{\left(t\right)}^{2}}{12.22}$$

The mean energy consumption during the above-mentioned stages is:(4)$$W_{TRS}=\overset{t}{\underset{0}{\int }}p{\left(t\right)}_{TRS}\cdot dt=\;\overset{N}{\underset{j=1}{\sum }}p_{j}\cdot \Delta t,$$
where $\Delta t=\frac{1}{f_{S}}\left[s\right]=4\;µs.\;$ The sampling rate is f_s_ = 250 ksps and the mean consumed power is given by:(5)$$p_{j}=\frac{p_{j}+p_{j+1}}{2}.$$

## 7. Methodology for Sizing Hybrid Storage Systems and Optimization

The two key aspects that should be considered for sizing the fast release storage element (super-capacitor) are: The input energy sources variation, and second, the inherent variation of the load of the AWS power supply that results from the variation of the data payload transmitted. The communication protocol has an important influence. Battery reliability and life span are affected by the rapid variations in current resulting from the transceivers operation. It is desirable to satisfy these short-term energy demands by a more reliable storage device like the super-capacitors to avoid battery aging. If, in the case of a battery, cyclabillity can vary between several hundred and tens of thousands of charging/discharging cycles, then in the case of super-capacitors, the number of cycles can reach or exceed one million cycles. The system proposed will keep the power demand constant, protecting the batteries and ensuring an appropriate power flow. It is essential to obtain the proper sizing of the hybrid source components; hence, the methodology for designing the power source must identify the functioning phases of the BT transceiver in order to capture the variation of the energy levels.

We propose a hybrid storage solution based on batteries and super-capacitors, as illustrated in Figure 20. The design commences with the precise identification of the transceiver’s power needs. The system will automatically switch between battery and super-capacitors.

Starting from the time diagram in Figure 21, we have determined the equivalent transceiver series resistance as a function of time R_T_(t) (R_trs_ in Figure 21). The time intervals are: T1:[0,t_1_] and T4:[t_3_,t_4_], active maximum level (0.5–0.6 V);T2: [t_1_,t_2_], intermediary level (wake up/sleep = 0.3 V);T3: [t_2_,t_3_], active minimum level (0.15–0.2 V), where t_1_ = 950µs, t_2_ = 1150µs, t_3_ = 1850µs, and t_4_ = 2400µs, as seen in Table 8.

In Figure 21, the equivalent series resistance of transceiver R_trs_ is also depicted as R_T_(t). For the K switches 1 means closed, 0 open, hence, theoretically, the control of the switches must implement the sequence shown in Figure 21. We determined the minimum capacitance of the super-capacitor and minimum battery capacitance that can assure the battery consumption smoothing.

The hypotheses considered are:(a)The transceiver operates usually in the voltage (supply) interval: [V_min_,V_max_], where V_min_ = 1.6 V and V_max_ = 3.6 V. We consider the variation of the supply voltage (for the voltage windows interval) as half of V_max_ (= 1/2 × 3.6 V = 1.8 V). Therefore, the new levels are: V_min_ = 1.8 V and V_max_ = 3.6 V.(b)For Li-Ion batteries the voltage window is [3.6 V,4.2 V], where 3.6 V represents SoC = 0%, and 4.2 V represents SoC = 100%. (SoC = state of charge).(c)We consider $R_{{\mathrm{SW}}_{\mathrm{ON}}}=0.3\;Ω$ for the analogue switch. The control voltage interval is [1.6 V,3.6 V]

The equivalent electric circuits of the power network of the wireless sensor for different time intervals corresponding to a specific period of BT transceiver transmission are illustrated in Figure 22.

In all the models, we have considered the battery as an ideal voltage source in the series with the internal resistance. Additionally, we have excluded the transitory regimes from the electrical analysis. For the sizing process, we consider the energy stored by the super-capacitor as ${\;W}_{e_{\mathrm{SC}}}$ for the [0,t_1_] interval. The energy consumed by the BT transceiver is W_e1_. The minimum necessary energy provided to the BT transceiver represents at least 75% of the energy provided by the super-capacitor, in order to keep the super-capacitor voltage variation between 100% and 50%, therefore:(6)$$W_{e1}\geq \frac{75}{100}\times W_{e_{SC}}$$
(7)$$W_{e_{SC_{min}}}=1.33\times W_{e1}$$

Knowing that
(8)$$W_{e_{SC}}=\frac{1}{2}\times {\mathrm{CU}}^{2}$$

We are now able to determine the capacitance value of the super-capacitor, starting from:(9)$$\frac{1}{2}\times C_{\mathrm{equiv}}\left(V_{\max}^{2}-V_{\min}^{2}\right)=1.33\times W_{e1}$$

Thus, we obtain Equation (10):(10)$${\;C}_{\mathrm{equiv}}=\frac{2.66\times W_{e1}}{V_{\max}^{2}-V_{\min}^{2}}$$

The constraint battery design is illustrated in Figure 22d. Having determined C_equiv_, we should verify that, during the load consumption period for the transceiver: [t_2_,t_3_] the battery is able to recharge the super-capacitor at the maximum voltage level. This is a mandatory constraint for assuring power continuity to the transceiver:$$W_{\mathrm{SC}}\leftarrow W_{B}$$

Zero condition:(11)$${\;U}_{\mathrm{SC}}=\frac{1}{2}V_{\max}\mathrm{or}\;V_{\min}$$

In this case, the limit current through the super-capacitor is illustrated in Figure 22e:(12)$$i_{SC}\left(t\right)=\left(1-\frac{R_{ESR_{SC}}}{R_{I_{B}}\left(t_{3}\right)}\right)\times i_{B}$$
where:(13)$${\;R}_{\mathrm{IB}}+R_{{\mathrm{ESR}}_{\mathrm{SC}}}+R_{{\mathrm{SW}}_{\mathrm{ON}}}=R_{E}$$

For validating the design, *i^*^_sc_* must be chosen so that the operation condition becomes:(14)$$V_{\mathrm{SC}}\left(t_{2},t_{3}\right)\geq V_{{\mathrm{SC}}_{\max}}$$
(15)$$V_{\mathrm{SC}}=V_{SC}\left(t_{2}\right)+\frac{1}{R_{E}C_{\mathrm{equiv}}}\overset{t_{3}}{\underset{t_{2}}{\int }}i_{\mathrm{SC}}\left(\tau \right)d\tau$$

There are two possible situations:If $V_{\mathrm{SC}}\left(t_{3}\right)\geq V_{\max}$ then operation based only on battery is sufficientIf $V_{\mathrm{SC}}\left(t_{3}\right)<V_{\max}$ then operation based only on battery is not sufficient, and $R_{\mathrm{IB}}^{\mathrm{new}}>R_{\mathrm{IB}}^{\mathrm{initial}}$. The calculated value for *C_equiv_* reaches 0.6 µF.

The above exhaustive study allows us to make several assumptions and remarks. The dependencies factors that must be considered are:The maximum data flow of transceiver in accordance with the process or applications requirements;The actual palette of BT implementations that can satisfy a large variety of applications;The environmental conditions that can play a significant role on design strategies.

The hybridization analysis proves that the increase in energy efficiency of the whole system is done in parallel with the improvement of reliability and life span. It is important to rationalize the communication session within the periods when the sensors are placed in low power or wake-up/sleep modes. Thus, for high data payloads, JDY-30 working in BT-EDR mode can be recommended for energy efficient communications and BLE HM-10 is useful for IoT type applications. The latter transceiver is more adequate for spontaneous telegram-based protocols.

The hybridization of the power supply/storage still remains of interest for all transceivers. The switches K1, K2, and K3 could be integrated on chip. A possible implementation could be based on graphene capacitors placed on chip. The use of hybrid supplies will smooth out the spikes or glitches resulted during transceiver operation. The comparison between the theoretical consumption curves and the real curves obtained in the case of low-cost transceivers reveals significant differences in energy mainly induced by the spikes. These are caused by inexpensive transceivers’ operation.

## 8. IoT Applications

As discussed in the second chapter, there are a lot of IoT-based implementations that can address various society needs. Yet, most of them depend on commercial platforms and transceivers, such as Xbee, MicaZ [105], and ESP8266 and development boards, such as Arduino and Raspberry Pi for implementing WSNs. New implementations based on recent standards for Bluetooth for short range to medium range communications and on nRF for longer range (up to 1 km) are addressed in this paper, in order to build autonomous wireless sensors with low costs, which are energy-efficient and that can adapt better to applications’ requirements, without depending on the limitations of previous or recent commercial solutions. We describe, in the following paragraphs, two such applications.

### 8.1. 3D Thermal (+Other Parameters: Humidity, Light etc.) Maps

BT/BLE based sensors can be deployed in many indoor environments where they collect data for large periods of time. As an example, thermal imaging can benefit from such IoT-based networks by providing a web-based 3D environment for temperature maps in commercial or residential buildings [106], as shown in Figure 23a. In order to reduce the costs, low-cost transceivers such as HC-05 (around 3 euro) and low-cost sensors such as DHT-22 (4 euro) are connected to an Arduino Board (from 3 to 7 euro) and can provide such 3D volumetric representations via Matlab. A solution, based on 3 AWSs, can reduce even more the costs, by using only 3 HC-05 transceivers instead of 20 or 24 such transceivers and employing intelligent data generation methods [107].

Additionally, lower cost sensors such as TMP-36z (less than 1 euro) and H5U21D (3 euro) can be used and provide the same accuracy: 2 Celsius degrees. On the other hand, thermal images can be also obtained with Fluke Ti20, which allows a much more precise representation but not a higher accuracy (the same: 2 Celsius degrees). Additionally, the costs are much greater (1300 euro). Further, thermal imaging via Fluke Ti20 provides only a snapshot of the temperature values at a given time, as illustrated in Figure 23b and additional calibration and software.Another aspect to be considered is the number of points used: Figure 23a requires only 20 or 24 points of measurement, while Figure 23b uses 12,288 points for thermal imaging. The Table 9 resumes all these observations:

### 8.2. RFM Based Application

The following application solves the problem of designing a large size solar power plant for monitoring and optimization of energy generation efficiency. It implements a data acquisition system using a dual wireless transceiver monitoring node. This application provides instantaneous signal acquisition: Current/voltage, panel temperature, by-pass diode activation periods, and based on these signals, computes the energy produced by each PV panel. The 3-tier architecture is composed of the lower level (i.e., collecting data from the PV panel in the vicinity of a local node) using BT-BLE transceivers organized in a “star” topology. The data aggregator at the middle-tier collects data from the local aggregators. The higher level represents the PV plant manager. It uses the polling method to control the data acquisition in order to provide the necessary information to the solar power plant.

The middle-tier topology depends on the terrain configuration, the size of the power plant and the implementation costs. The block diagram of the wireless nodes is illustrated in Figure 24:

The software that implements the acquisition algorithm assures asynchronous signal monitoring. The two processes are illustrated in Figure 25. The first data flow is synchronous with the signal acquisition sampling rate. All these signals are converted from analogue to digital and stored into the shared memory.

This structure allows the implementation of local control functions, it is also able to redirect or simply transmit data collected to the supervisor.

## 9. Conclusions and Future Work

This research enables a better understanding of the AWS novel architecture for the fast release supply, based on super-capacitors. We investigated and quantified several aspects related to information flow and energy efficiency. As a result, information heterogeneity and the power supplied will influence the life span, reliability, and availability of the AWS.Thus, the insertion of super-capacitor elements generates improvements in energy efficiency by smoothing instantaneous parameters (current, voltage, and power). From the power supply perspective, energy harvesting from the surrounding environment is possible. The fast storage element must provide power continuity to the AWS. On the other hand, the transfer of information generates significant variations of the power supply load that can be compensated using super-capacitors. The benefits consist in battery stress avoidance, and leads to a decrease in internal power supply resistance, improving the transceiver’s communication. Details of these aspects will be analyzed in future work. Additionally, we plan to investigate the integration of fast storage elements inside the transceiver by considering technological advances in the area of condensed matter.

Field tests led to the following observations:There are dependencies between different data payload flows (with command, from 50 to 500 characters), stages (disconnected, no command/command), modes (echo/no echo), and distance between transceivers and transceiver type.There is a difference in power consumption, from 7% to 30%,for data payload content at extremes (null vs. “U” characters), for the actual transmission period (2.5 ms),Energy efficiency can be optimized by taking into account the above observations.

The results confirm that the power supply “hybridization” (super-capacitors and batteries), as well as the communication “hybridization” (using two different transceiver types) generates an improvement in energy efficiency. The communication protocol will adapt to the amount of data transferred. The communication parameters (e.g., range, data payload) will affect the overall energy efficiency and the system reliability, hence they must also be considered in the AWS design.

The developed methodology enables optimal super-capacitor sizing.The parameters related to the emission field and data exchanged were correlated with the energy consumption and transceiver type in order to adjust the sensors’ parameters, e.g., sensor placement, power tuning, communication protocol, and transceiver type. For the new BLE standards, the proposed design strategies can be adapted based on the examples analyzed herein.

## Figures and Tables

**Figure 1 sensors-19-03364-f001:**
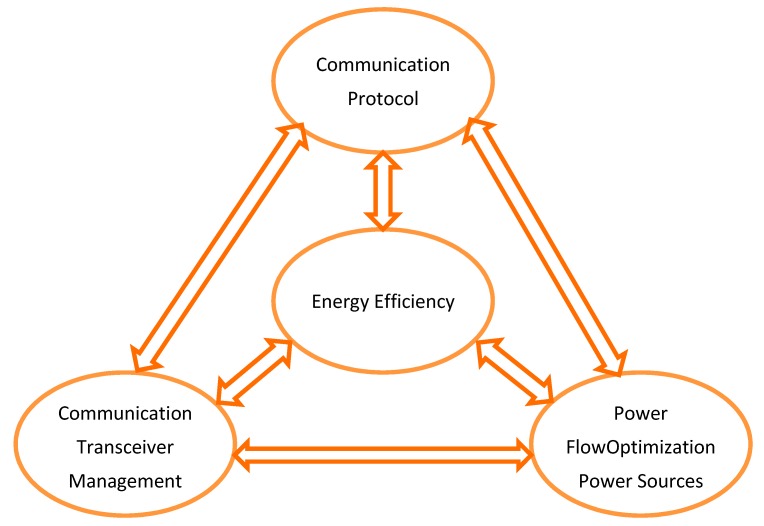
Autonomous Wireless Sensors (AWS) design and energy efficiency.

**Figure 2 sensors-19-03364-f002:**
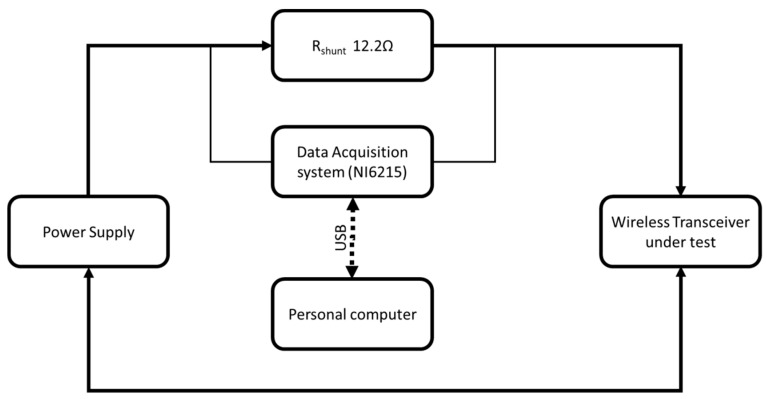
Setup diagram (low resistance = 12.22 Ω).

**Figure 3 sensors-19-03364-f003:**
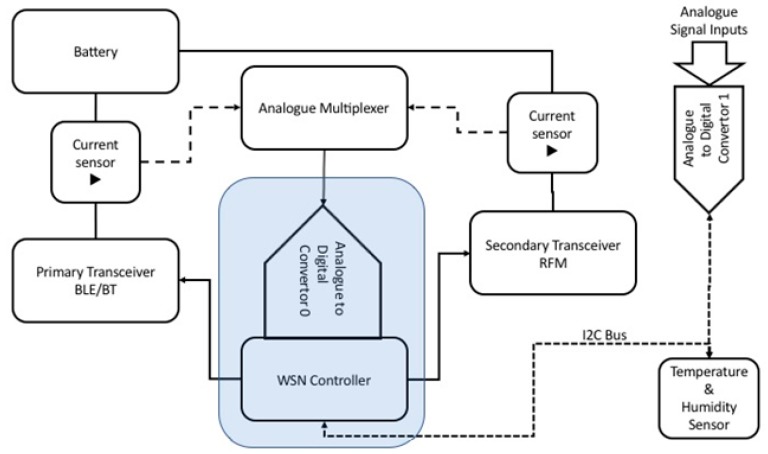
Block diagram of the experimental AWS.

**Figure 4 sensors-19-03364-f004:**
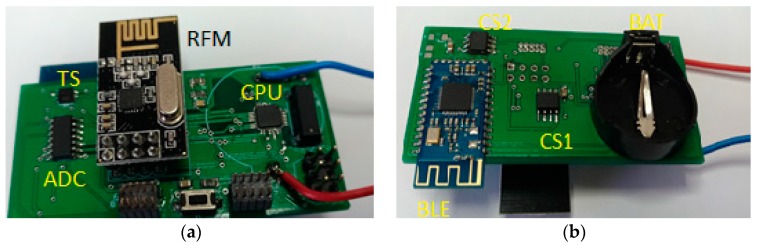
AWS: (**a**) Top layer (CPU-ATMega88; RFM transceiver-NRF24 L01, 2.4 GHz; ADC-MCP3428; Temperature Sensor-HTU21D); (**b**) bottom layer: BLE transceiver—HM10BLE; CS1/CS2-current sensors (ACS 712-05) based on the Hall Effect.

**Figure 5 sensors-19-03364-f005:**
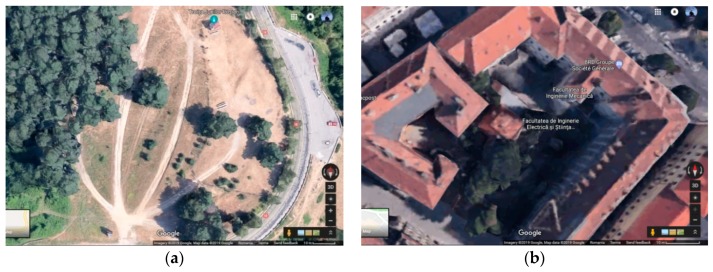
Test scenarios: (**a**) Troita Junilor park (Brasov periphery); (**b**) NII2 building (Brasov center).

**Figure 6 sensors-19-03364-f006:**
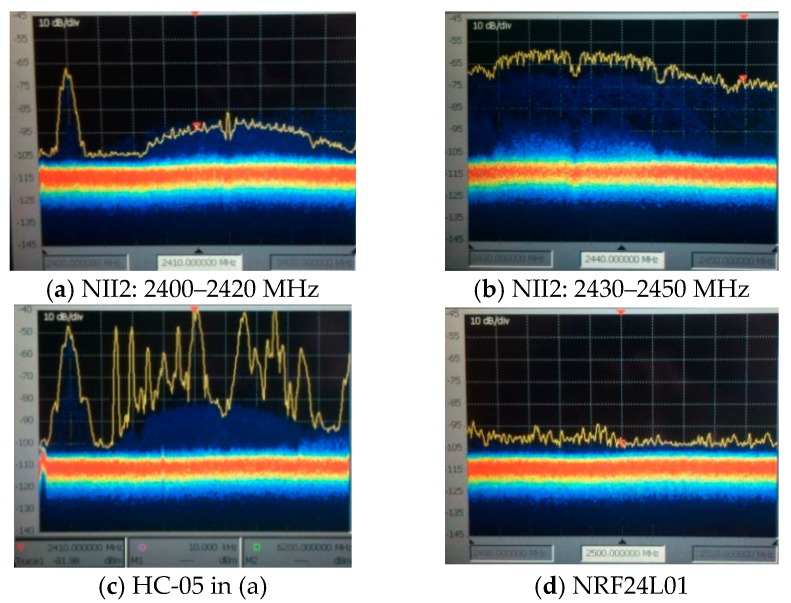
A room inside the NII2 building. Spectrum bands: (**a**) Semi-occupied spectrum; (**b**)occupied spectrum; (**c**) HC-05 in 2400–2420 MHz band; (**d**) ISM outer band (unoccupied spectrum), for NRF24L01.

**Figure 7 sensors-19-03364-f007:**
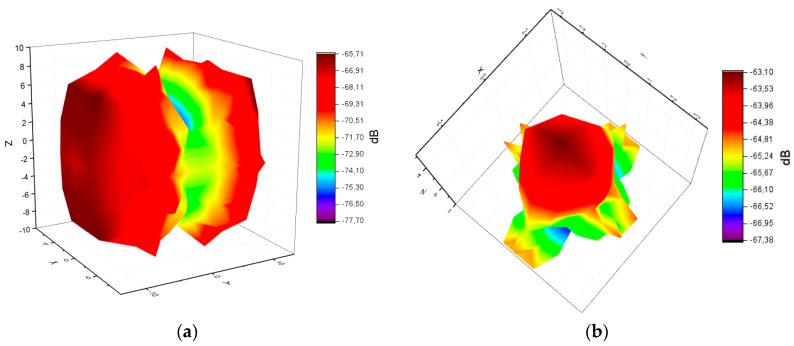
Measured: (**a**) MicaZ antenna emission field (dB); (**b**) HC-05 antenna emission field.

**Figure 8 sensors-19-03364-f008:**
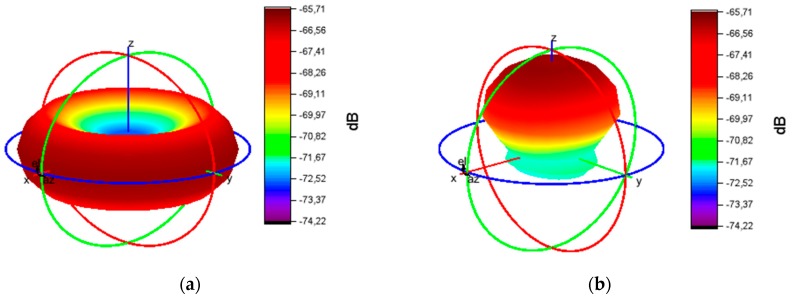
Theoretical: (**a**) MicaZ antenna emission field (dB); (**b**) HC-05 antenna emission field.

**Figure 9 sensors-19-03364-f009:**
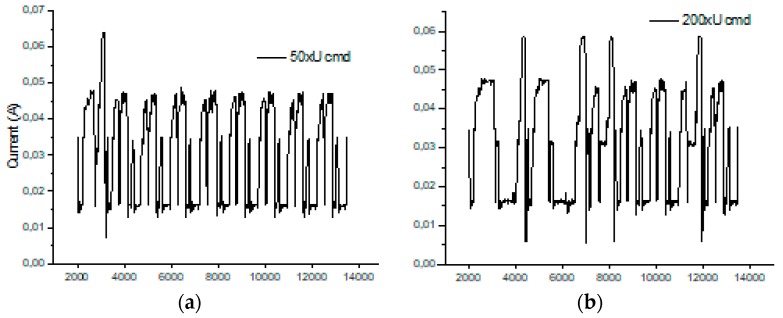
HC-05 Current over 12ms for a 12.2Ω resistor (current sensor resistance)(no echo mode): (**a**) a command of 50 U (55H) characters sent in burst; (**b**) a command of 200 U (55H) characters sent in burst; (**c**) a command of 500 U (55H) characters sent in burst; (**d**) no command was sent.

**Figure 10 sensors-19-03364-f010:**
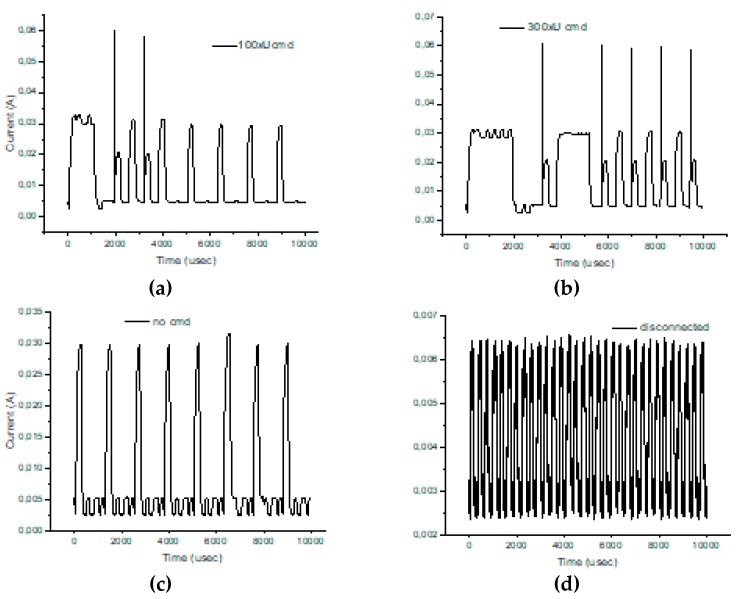
JDY-30 Current over 10ms for a 12.2Ω resistor (no echo mode): (**a**) a command of 100 U (55H) characters sent in burst; (**b**) a command of 300 U (55H) characters sent in burst; (**c)** no command was sent; (**d**) in disconnected state.

**Figure 11 sensors-19-03364-f011:**
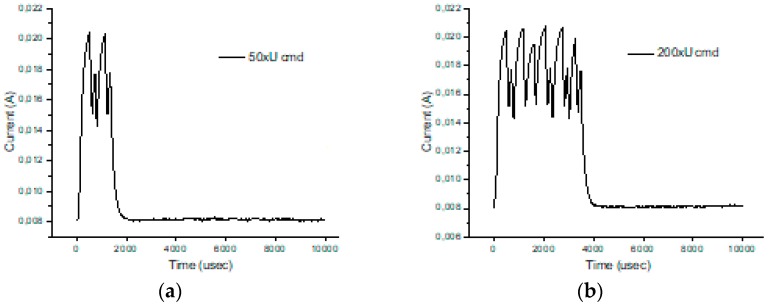
HM-10 Current over 10 ms for a 12.2Ω resistor (no echo mode): (**a**) a command of 50 U (55H) characters sent in burst; (**b**) a command of 200 U (55H) characters sent in burst; (**c)** no command was sent; (**d**) in disconnected state.

**Figure 12 sensors-19-03364-f012:**
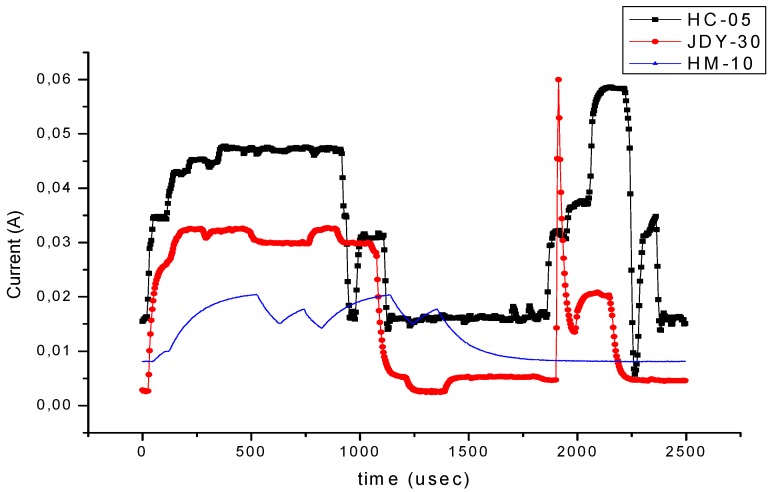
Variation for three transceivers: HC-05, JDY-30 and HM-10 during a 2.5 ms transmission period (echo mode).

**Figure 13 sensors-19-03364-f013:**
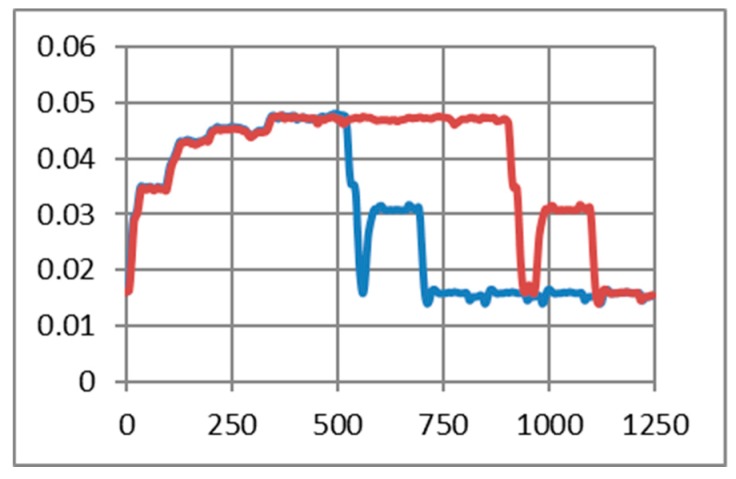
HC-05 current consumption [A] waveform when “U” characters were transmitted (50 vs. 100) for 1.25 ms. (The spikes were cleaned) (no echo mode).

**Figure 14 sensors-19-03364-f014:**
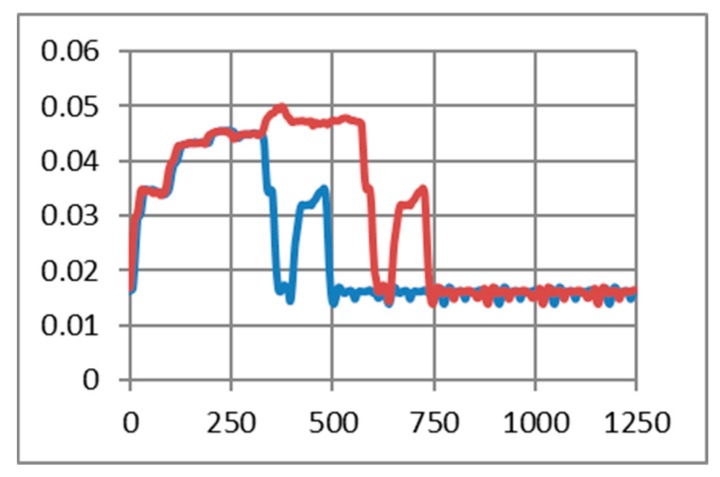
HC-05 current consumption [A] waveform when null characters were transmitted (50 vs. 100) for 1.25 ms. (The spikes were cleaned) (no echo mode).

**Figure 15 sensors-19-03364-f015:**
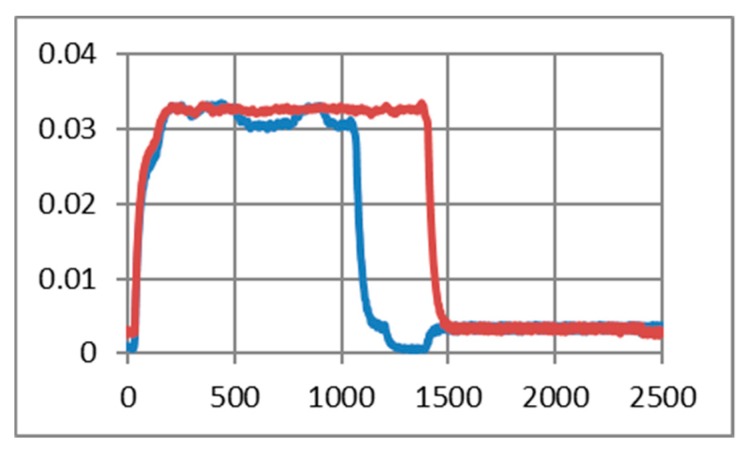
JDY-30 current consumption [A] waveform when null characters were transmitted (50 vs. 100) for 2.5 ms. (The spikes were cleaned) (echo mode).

**Figure 16 sensors-19-03364-f016:**
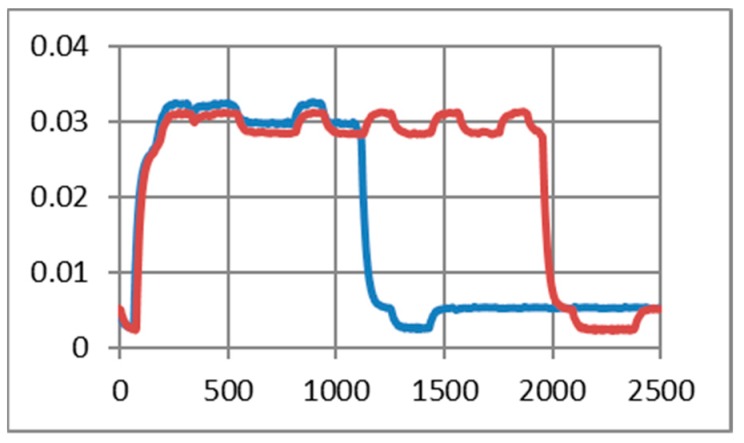
JDY-30 current consumption [A] waveform when “U” characters were transmitted (50 vs. 100) for 2.5 ms. (The spikes were cleaned) (echo mode).

**Figure 17 sensors-19-03364-f017:**
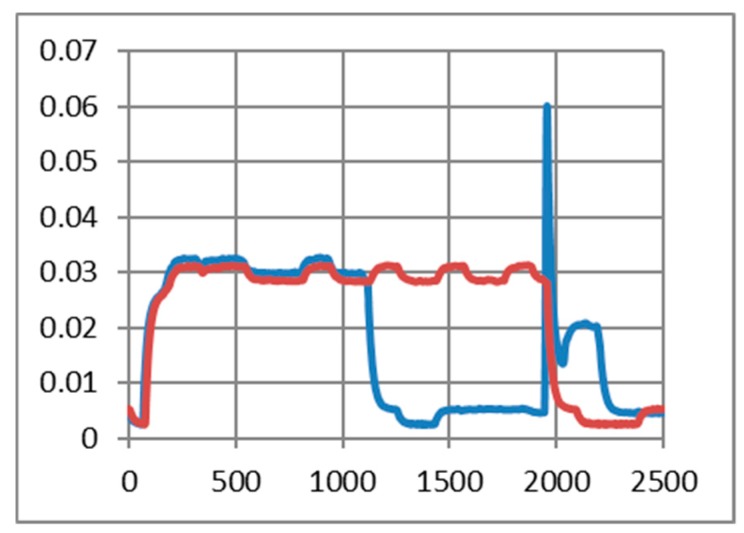
JDY-30 current consumption with the inherent spike (50 vs. 100 U transmitted characters) (echo mode).

**Figure 18 sensors-19-03364-f018:**
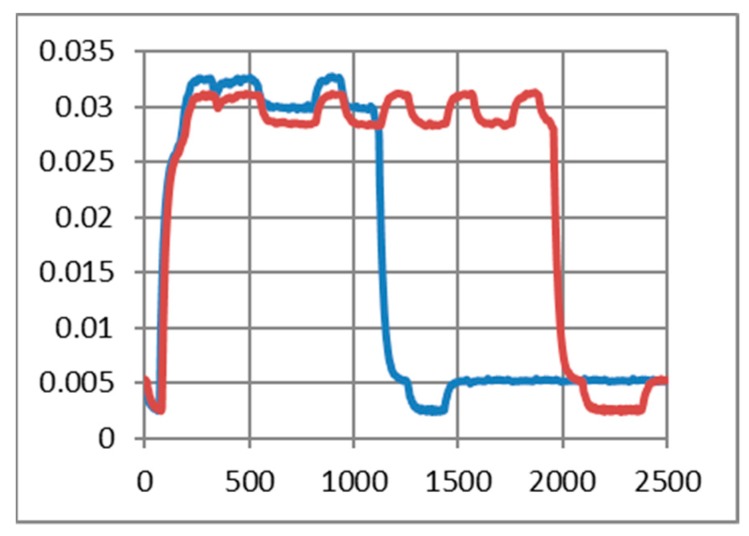
JDY-30 current consumption without the spike (50 vs. 100 U transmitted characters) (echo mode).

**Figure 19 sensors-19-03364-f019:**
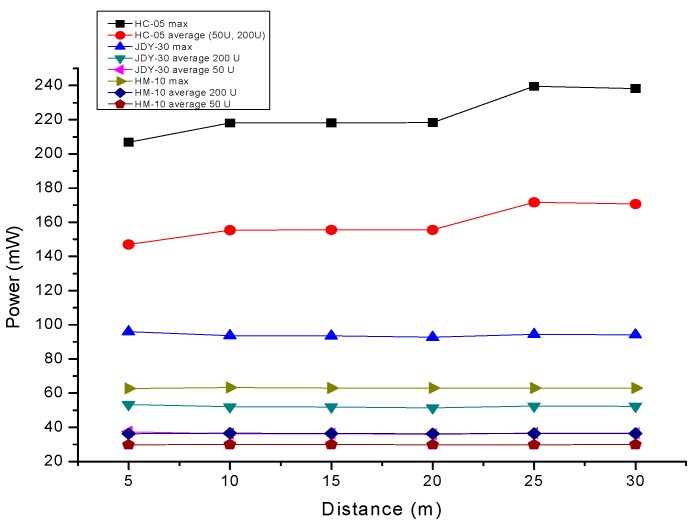
Power drawn by the HM-10, JDY-30, and HC-05 transceivers depending on distance.

**Figure 20 sensors-19-03364-f020:**
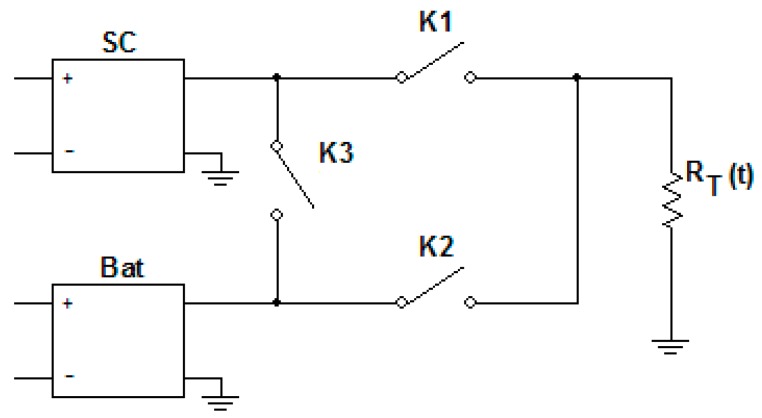
Storage system circuit.

**Figure 21 sensors-19-03364-f021:**
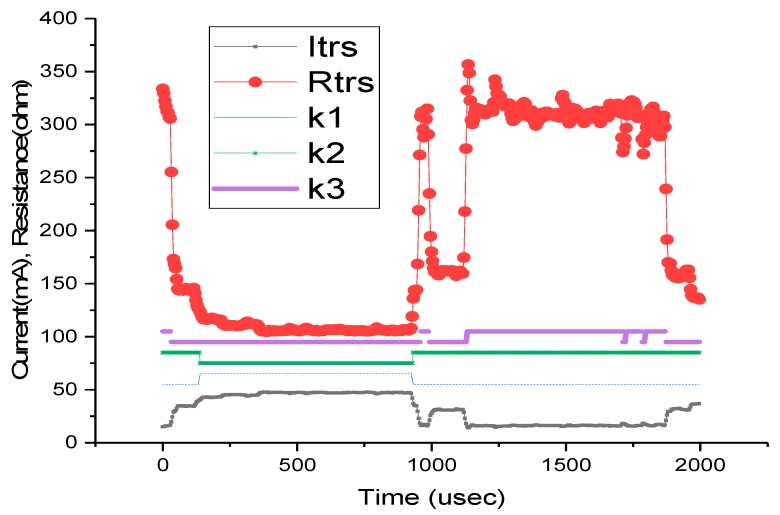
Time Diagrams for HC-05: for four successive time intervals.

**Figure 22 sensors-19-03364-f022:**
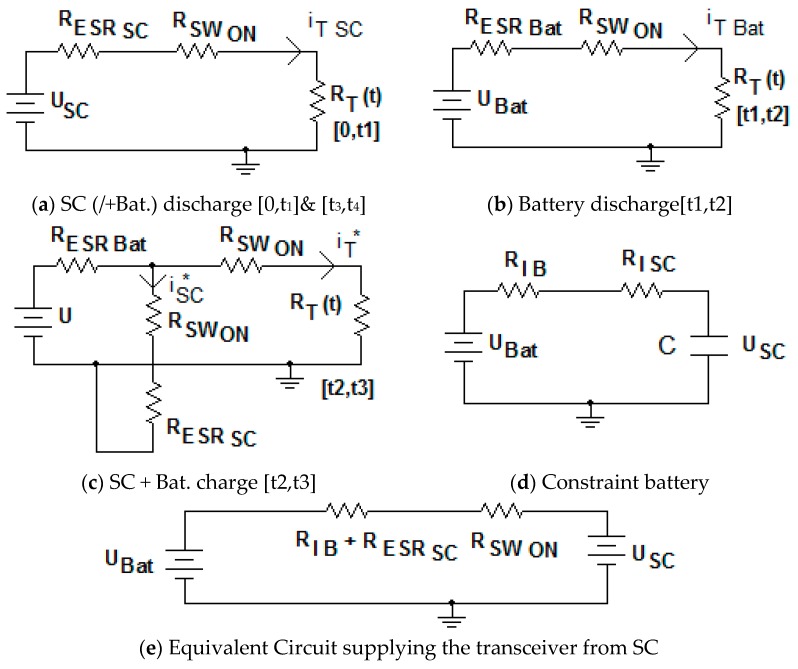
Electric circuits.

**Figure 23 sensors-19-03364-f023:**
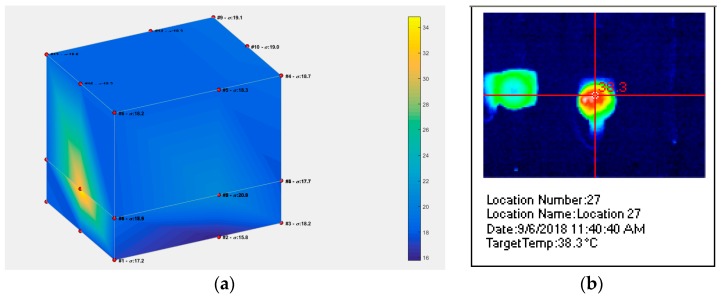
Volumetric representations: (**a**) IoT-based, via 12 or 20 sensors connected to HC-05 transceivers, (**b**) based on Fluke IR thermal imaging.

**Figure 24 sensors-19-03364-f024:**
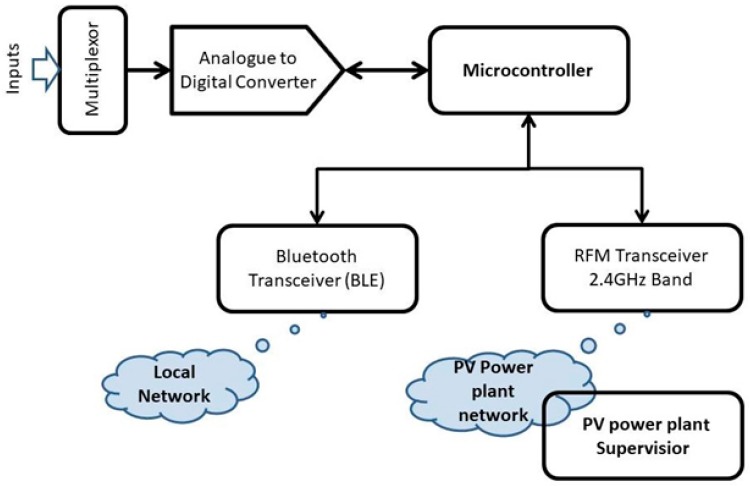
Block diagram of the hybrid DAQ node integrated into a signal acquisition network.

**Figure 25 sensors-19-03364-f025:**
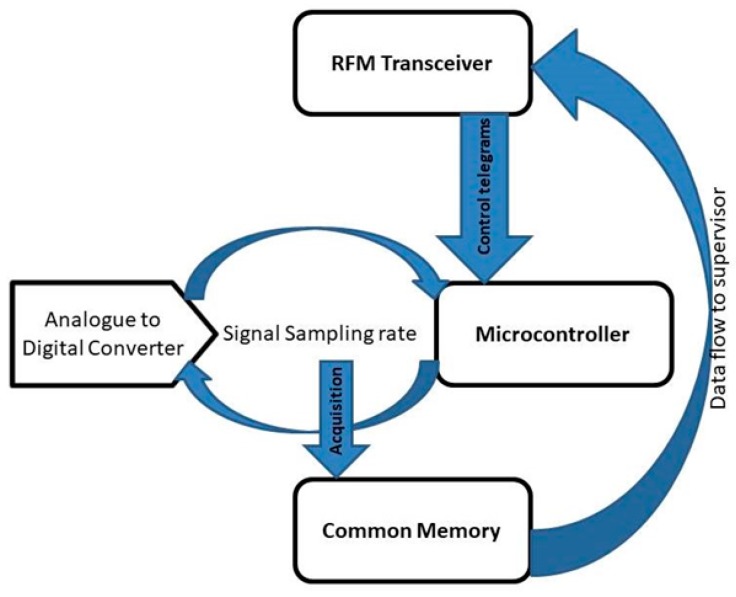
Data flows in the system.

**Table 1 sensors-19-03364-t001:** Synthesis of AWS Technologies and Standards [4,5,6,7].

	Short to Medium Range			Long Range		Proprietary
**Metric per technology**	ZigBee/ *802.15.4e	Bluetooth/ *BLE	Wi-Fi/ *802.11ah	LoRa	MIOTY	nRF24
	**Radio Spectrum Performance**					
**Main Freq. bands**	868/915 MHz & 2.4 GHz	2.4 GHz	2.4–5 GHz/770, 868,915 MHz	868/915 MHz	868 MHz	2.4 GHz
**Spreading sequence & Ch. bandwidth**	DSSS/+TSCH	FHSS	MC-DSSS, CCK	CSS	CSS	DSSS
	2 MHz	1 MHz	22 MHz/1–16 MHz	<500 KHz	200 KHz	1 MHz
**RF channels &IF band resist.**	1,10 & 16	79/40	11 to 24	10 EU, 8US	unknown	126
	modest	good	best	good	best	poor
	**Power Consumption**					
**Sleep & Peak current**	4.18 μA	0.78 μA	50–70/≅1 μA	1 μA	1 μA–10 μA	26 μA
	30–40 mA	30/15 mA	116/22 mA	32 mA	unknown	18 mA
**Power cons. watts**	low	med/low	high/low	low	low	low
	36.9 mW	215 mW/10 mW	835 mW/≅200 mW	100 mW	unknown	60 mW
**Pow. Efficiency**	0.15 μW/bit	186 μW/bit	0.005/50 μW/bit	1.5 μW/bit	unknown	2.48 μW/bit
	**Data Flow**					
**Data rate &Max. throughput**	250 Kbps	1–25 Mbps/3 Mbps	11,54,300 Mbps 0.15–346 Mbps	50 Kbps	0.4 Kbps	2 Mbps
	150 Kbps	2 Mbps/300 Kbps	7,25,100 Mbps/≅40 Mbps	22 Kbps	unknown	372–512 Kbps
**Latency**	20–30 ms	100 ms/6 ms	50 ms/≅1 ms	>1 s	unknown	20–30 ms
**Coverage**	10–300 m	10–30 m/10 m	100–500 m/1 km	5 km	<15 km	10–50 m
**Connectivity**	Possible w. 6 LP	yes	yes	Possible w. 6LP	Possible, no IP cnct.	yes (limited)

**Table 2 sensors-19-03364-t002:** Lower cost IoT-based devices (BLE, Wi-Fi) represent a solution to ZigBee candidates.

WSN IoTDevelopment Platforms and Modules								
**Transceiver**	ESP8266	NRF24L01	HM-10	HC-05	AMS001/002	LM811	MicaZ	Xbee
**Standard**	Wi-Fib/g/n	Nrf24	BLE	BT	BLE	BLE/Wi-Fi	ZigBee	ZigBee
**Supply**	3.3 V	3.3 V	2–3.7 V	3.6–6 V	1.8–3.6 V	3.3/5 V	2.7–3.3 V	3.3 V
**Current draw** **Tx and Rx**	100–150 mA	Tx 7–11.3Ma Rx 9–13.5 mA	8.5–9 mA	~30 mA	Tx 13/23 mA Rx 11/25 mA	150 mA	Tx17.5 mA Rx19.7 mA	Tx 45 mA Rx 50 mA
**Max range**	100 m	10–50 m	10–20 m	10–20 m	10–20 m	10–20 m	20–70 m	10–100 m
**Size (mm)** **Weight(g)**	(10–18) × (20–24), 2–20 g	(12–18) × (18–40), 10–20 g	13 × 27, 8 g	(13–15) × (27–28), 15–20 g	11.4 × 17.6, 20 g	12 × 25, 25 g	32 × 58, 20 g	23× (27–33), 40–70 g
**Cost**	5–10$	≅5$	5–10$	≅5$	5–10$	10–20$	300$	30–200$

**Table 3 sensors-19-03364-t003:** Range, throughput, and interference characteristics for HC05, JDY-30, HM-10, and nRF24.

Characteristic	HC-05	JDY-30	HM-10	NRF24
Indoor scenario range (same floor level)	10–15 m	10 m	5 m	15–25 m, 100–200 m ^1^
(between floors)	6–10 m	5 m	2–3 m	15 m, 100 m ^1^
Throughput loss under interference	Severe: 30–50%, Average: 15–20%	Severe: 45–60%, Average: 20%	Severe: 70–80%, Average: 25%	Not higher than 20%
Indoor scenario in-band interference ^2^	considerable	considerable	worst effect	negligible
Outdoor scenario range	30–40 m	30 m	20–30 m	100 m, 1 km ^1^

^1^ Only possible by adding a long range antenna (gain: 20–30 dB). ^2^ Based on range and throughput loss measurements and estimations [78,79,82].

**Table 4 sensors-19-03364-t004:** Comparison between theoretical models and models based on measurements, see Figure 7 and Figure 8.

Characteristic	MicaZ	HC-05
Theoretical model: dBm range	−65.71 to −74.22	−65.71 to −74.22
Model based on measurements: dBm range	−65.71 to −77.70	−63.1 to −67.38
Difference in dBm between theoretical model and model based on measurements	Average: 0.5, Maximum: 3.48	Average: 0.9, Maximum: 6.86

**Table 5 sensors-19-03364-t005:** Current consumption for BT transceivers (voltage supply 3.3 V for HM-10, JDY-30, and 5 V for HC-05).

I(mA)	Tx&Rx	Mean	Spike	Nocmd
**BLE [100]**	17.5	8.53	16	7.4
**HM-10**	20.60	10.47	18 ^1^	8.80
**JDY-30**	31.98	14.40	60.43	8.53
**HC-05**	47.25	31.53	62.02	18.14

^1^ the spike is not outside the transmission period, as opposite to Reference [100].

**Table 6 sensors-19-03364-t006:** Current ratio illustrating the deviations from standard BLE transceiver and the commercial transceivers used in the experiments.

%	Tx&Rx	Mean	Spike	Nocmd
**BLE [100]**	100	100	100	100
**HM-10**	117.71	122.77	112.5 ^1^	118.88
**JDY-30**	182.74	168.86	377.69	115.22
**HC-05**	269.99	369.64	387.62	245.15

^1^ the spike is not outside the transmission period, as opposite to [100].

**Table 7 sensors-19-03364-t007:** Energy consumption for the three tested transceivers.

**With Spikes**	**Energy [µJ]/2.5 ms**				**Energy/Byte [nJ/char]**			
	**50 U**	**50 Null**	**100 U**	**100 Null**	**50 U**	**50 Null**	**100 U**	**100 Null**
**HC-05**	391.23	431.78	447.19	442.79	7.133	7.158	3.806	3.897
**JDY-30**	129.19	122.20	170.47	154.39	3.188	3.352	1.572	1.681
**HM-10 ^1^**	103.07	106.73	132.51	125.43	1.916	1.989	1.060	1.029
**Without Spikes**	**Energy [µJ]/2.5 ms**				**50**		**100**	
	**50 U**	**50 Null**	**100 U**	**100 Null**	**U**	**Null**	**U**	**Null**
**HC-05**	337.82	328.80	406.41	345.41	372%	360%	359%	379%
**JDY-30**	116.70	108.19	170.47	140.65	166%	169%	148%	163%
**HM-10 ^1^**	103.07	106.73	132.51	125.43	100%	100%	100%	100%

^1^ For HM-10 the spikes are integrated within the transmission period (inside it).

**Table 8 sensors-19-03364-t008:** Consumption [µJ] in our experimental settlement for HC05, JDY-30, and HM-10 for 2500 µs.

*T1*	*T2*	*T3*	*T4*
204,703	30,205	55,683	106,703
94,484	0 ^1^	19,219	15,490 ^1^
80,501 ^2^	0 ^2^	22,574	0 ^2^

^1^ The *T2*: [t_1_,t_2_] interval can be replaced with the *T4*: [t_3_,t_4_] interval due to reduced energy and period. ^2^ The spike interval is considered as part of the *T1*: [0,t_1_] interval since it is not outside the transmission period. Additionally, because its peak is lower than the transmission level (around 20%–30% less) and its period is much shorter (70%–80% less), it could be added to the *T2*: [t_1_,t_2_] interval. See Figure 12.

**Table 9 sensors-19-03364-t009:** Solutions for displaying temperature distribution on small surfaces (few meters).

Solution	Real-Time	Points of Representation	Cost	2D or 3D, No. of Bits, Temp. Accuracy (°C)
Fluke TI 20	yes	12,288	1300 euro	2D, 14 bits, 2 °C
12 HC-05s+12 DHT22s with Arduino	no, 2 sets of measurements	12 × 2 = 24	90 euro	3D, 8 bits, 2 °C
20 HC-05s+20 DHT22s with Arduino	yes	20	150 euro	3D, 8 bits, 2 °C
Our solution with 3 AWSs	yes	20–24	30–90 euro ^1^	3D, 10–14 bits, 2 °C

^1^ lowest cost = 30 euro: each AWS has 8 TMP36z (10 bits) sensors (the price of AWS is around 10 euro), highest cost = 90 euro: each AWS has 8 H5U21D (14 bits) sensors (the price of AWS is around 30 euro).

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
