# Peer review of "Design of Wireless Sensors for IoT with Energy Storage and Communication Channel Heterogeneity"

_sensors, 2019, doi:10.3390/s19153364_

Reviewer 1 Report

The authors have identified a topic worthwhile of study and presented decent results. While the paper is interesting, further clarification and study is required as listed below.

The title states the design of wireless sensors for IoT, but not much of the manuscript is focused on the wireless transceiver power analysis. It is suggested to change the title to better reflect the content.

The experimental study was conducted on BT and BLE with good findings. While these technologies are broadly adopted in IoT applications, they are limited in capabilities such as range of communication and networking. It is recommended that this study is performed on additional technologies such as nRF, Xbee, and LoRa for the other advantages they provide such as mesh networking and long-range-short-data communication.

While the power analysis is good, it is better to validate the efficiency of the hybrid storage system through implementation on an IoT application. The authors are recommended to implement an IoT application with and without the hybrid storage system using discrete components, and highlight the improvements obtained.

The study states that data payload causes the transceiver power consumption to vary, as it has been validated by several previous studies. It is recommended for the authors to conduct an extensive study and present the underlying trendlines (correlations) between data payloads and power consumption.

This manuscript has a weakness in the discussion section. While the quantitative results are presented, detailed qualitative analysis of results is missing. How are the findings applicable to IoT applications? How does the study shine light on communication channel heterogeneity? What future works are planned from the results obtained? 

Reviewer 2 Report

The paper presents a lot of information, especially the comparison tables. As the nature of the journal, open access, will provide readers with a lot of information. However, I feel, the authors cited " sensors" journal more times than other journals. The papers in SHM, NDT/NDE can provide more IOT, energy harvesting information. The authors can add a few more articles to highlight the importance of energy harvesting and SHM.

quality: ok

information: ok

need a few more citations. 

In page 5:

Authors mention “ WSNs are used in many domains e.g. military, industrial, environmental, residential, and health care. [here, authors can cite a couple more to draw the attention of other users of WSNs]

Depending on their requirements and sensor capabilities one can define WSNs in terms of size (small to very large scale), sensors’ capacity  (homogeneous to heterogeneous), topology and mobility (static, mobile and hybrid) [17]. [ here, authors can add examples, and cite a few articles]

WSNs communicate sensitive data, thus security concerns must be addressed at the beginning of the system design [16]. [ Authors should have more citations here, as this is the most important point]

In pages 7, 8

AWS Design and Implementation: this section is written without citations, [authors, try to define your parameters if possible, cite them]

In page, 9, 10

Band Coexistence for short to medium range communications: This is an important aspect, can be improved by adding a table, to discuss few more practical cases, are at least cite some articles where such applications were carried out in the past if any

Are there any assumptions involved in 3D Simulation and Visualization of the AWS Emission Fields and AWS’ power consumption? currents from 40 to 65mA? Reason for considering such range? Any citations here?

In page 19, 20

Why don’t you call, “Methodology for sizing hybrid storage systems & optimization” as “ optimization of energy harvesting system? Is it not similar to harvesting?

The paper is good, informative, write your assumptions in the works you carried out for repeatability purpose.

Reviewer 3 Report

  In this paper, a hybrid solution based on a novel architecture that duplicates the transceivers and also the power source represented by a hybrid storage system. By identifying the consumption needs of transceivers, an appropriate methodology for sizing and controlling the power flow for the power source is proposed.

 The paper is well written and easy to follow and the problem is worthy investigation.

 In the literature review, the authors are invited to discuss the following references regarding the resource allocation for better wireless energy harvesting in wireless networks.

 H. Tran et al , "Robust Design of AC Computing-Enabled Receiver Architecture for SWIPT Networks," in IEEE Wireless Communications Letters, vol. 8, no. 3, pp. 801-804, June 2019.
H. Tran, et al, "Resource Allocation in SWIPT Networks Under a Nonlinear Energy Harvesting Model: Power Efficiency, User Fairness, and Channel Nonreciprocity," in IEEE Transactions on Vehicular Technology, vol. 67, no. 9, pp. 8466-8480, Sept. 2018.

H. Tran et al, "RF Wireless Power Transfer: Regreening Future Networks," in IEEE Potentials, vol. 37, no. 2, pp. 35-41, March-April 2018.

Y. Liu, et all, "Intelligent Edge Computing for IoT-Based Energy Management in Smart Cities," in IEEE Network, vol. 33, no. 2, pp. 111-117, March/April 2019.
X. Shao, et all, "Dynamic IoT Device Clustering and Energy Management With Hybrid NOMA Systems," in IEEE Transactions on Industrial Informatics, vol. 14, no. 10, pp. 4622-4630, Oct. 2018.

 The authors are invited to discuss the impact of the power consumption to maintain a given BER. In addition the adaptive modulation can be discussed and its impact on power and transceiver design complexity (no need to add any extra measurement)

 The authors are invited to better discuss the future challenges and research paths of this work.

Author Response

Round  2

Reviewer 1 Report

The authors have done a good job of addressing the suggestions made. I would suggest an extensive proof reading of the manuscript prior to publication, as there are minor typographical and grammatical errors. For instance, the incorrect figure number in page 26.

Reviewer 3 Report

In my opignion, the paper is ready for publication.
